

# Establishment of a prognostic risk prediction modelfor non-small cell lung cancer patients with brainmetastases: a retrospective study

Fei Hou[1,*], Yan Hou[2,*], Xiao-Dan Sun[3], Jia lv[1], Hong-Mei Jiang[1], Meng Zhang[1], Chao Liu[1] and Zhi-Yong Deng[1]

[1] Department of Nuclear Medicine, Yunnan Cancer Hospital (The Third Affiliated Hospital of Kunming Medical University), Kunming, Yunnan, China
[2] Department of General Practice, China Medical University, Shenyang, Liaoning, China
[3] Department of Publicity, Yunnan Cancer Hospital (The Third Affiliated Hospital of Kunming Medical University), Kunming, Yunnan, China
[*] These authors contributed equally to this work.

Corresponding authors
Chao Liu, liuchao@kmmu.edu.cn
Zhi-Yong Deng,
13888158986@163.com

## ABSTRACT

**Background**. Patients with non-small cell lung cancer (NSCLC) who develop brain metastases (BM) have a poor prognosis. This study aimed to construct a clinical prediction model to determine the overall survival (OS) of NSCLC patients with BM.
**Methods**. A total of 300 NSCLC patients with BM at the Yunnan Cancer Centre were retrospectively analysed. The prediction model was constructed using the least absolute shrinkage and selection operator-Cox regression. The bootstrap sampling method was employed for internal validation. The performance of our prediction model was compared using recursive partitioning analysis (RPA), graded prognostic assessment (GPA), the update of the graded prognostic assessment for lung cancer using molecular markers (Lung-molGPA), the basic score for BM (BSBM), and tumour-lymph node-metastasis (TNM) staging.
**Results**. The prediction models comprising 15 predictors were constructed. The area under the curve (AUC) values for the 1-year, 3-year, and 5-year time-dependent receiver operating characteristic (curves) were 0.746 (0.678–0.814), 0.819 (0.761–0.877), and 0.865 (0.774–0.957), respectively. The bootstrap-corrected AUC values and Brier scores for the prediction model were 0.811 (0.638–0.950) and 0.123 (0.066-0.188), respectively. The time-dependent C-index indicated that our model exhibited significantly greater discrimination compared with RPA, GPA, Lung-molGPA, BSBM, and TNM staging. Similarly, the decision curve analysis demonstrated that our model displayed the widest range of thresholds and yielded the highest net benefit. Furthermore, the net reclassification improvement and integrated discrimination improvement analyses confirmed the enhanced predictive power of our prediction model. Finally, the risk subgroups identified by our prognostic model exhibited superior differentiation of patients' OS.
**Conclusion**. The clinical prediction model constructed by us shows promise in predicting OS for NSCLC patients with BM. Its predictability is superior compared with RPA, GPA, Lung-molGPA, BSBM, and TNM staging.

## INTRODUCTION

Lung cancer ranks among the most prevalent malignancies worldwide, with non-small cell lung cancer (NSCLC) being the predominant pathology (*Sung et al., 2020*). Interestingly, the brain is the most common site of distant metastasis in NSCLC. Approximately 10% of patients present with brain metastases (BM) at the time of diagnosis, while an additional 40–50% of patients develop BM during their disease (*Ouyang et al., 2020*). The prognosis of NSCLC with BM is extremely poor, with untreated patients having a median overall survival (OS) of only 1–3 months, and a 1-year survival rate of 10–20% (*Schuler et al., 2016*). Treatment modalities for NSCLC with BM include radiotherapy, surgery, chemotherapy, molecular targeting, and immunotherapy, among others, which can be categorised as local or systemic interventions. Given the diverse clinicopathological characteristics observed in patients, accurate prognosis prediction for NSCLC patients with BM is important for selecting a more individualised treatment strategy.

While the tumour-lymph node-metastasis (TNM) staging system remains the gold standard for assessing cancer prognosis, it is not without limitations. First, it is primarily based on the anatomical progression of the disease, assuming that more advanced staging corresponds to poorer prognostic outcomes. However, patients with identical anatomical progression might still exhibit varying prognoses. Second, TNM staging does not include the primary tumor size, lymph node metastasis, and distant metastasis as continuous variables, which can create an imprecise staging. Additionally, TMN staging does not take other variables into consideration to predict the patient prognosis, such as age, gender and histology that would contribute to predicting patient prognosis (*Balachandran et al., 2015*). Consequently, TNM staging proves inadequate in accurately predicting the prognosis of NSCLC patients with BM.

Currently, the most widely used prognostic models for BM are the recursive partitioning analysis (RPA), graded prognostic assessment (GPA), the update of the graded prognostic assessment for lung cancer using molecular markers (Lung-molGPA), and the basic score for BM (BSBM) (*Gaspar et al., 2000*; *Sperduto et al., 2008*; *Sperduto et al., 2017*; *Gao et al., 2018*). The RPA focuses on factors such as age, the Karnofsky performance score (KPS), control of the primary tumour, and the presence of extracranial metastases. GPA focuses on factors such as age, the KPS, the number of BM, and the presence of extracranial metastases. The Lung-molGPA, in addition to GPA factors, incorporates the mutation status of NSCLC driver genes. Lastly, the BSBM focuses on the KPS, control of the primary tumour, and the presence of extracranial metastases. While these models are simple and convenient to use, they have certain limitations. For instance, RPA and GPA are general models for BMs without a specific focus on lung cancer primaries, and BSBM does not consider brain metastatic lesion characteristics. The Lung-molGPA, being a specific model for lung cancer BM, stands as a comparatively robust model. However, these models employ subjective

or challenging to quantify evaluation metrics. Furthermore, only a few researchers have developed prognostic prediction models for BM in patients with NSCLC. For instance, *Li et al. (2022)* introduced a novel prognostic model based on clinical features and inflammation markers to enhance the prognostic information accuracy for NSCLC patients with BM compared with the adjusted prognostic analysis, RPA, and GPA. Additionally, *Zhang et al. (2020)* examined the feasibility of employing computed tomography imaging radiomics to predict the survival of NSCLC patients with BM undergoing whole-brain radiotherapy.

However, these studies have certain limitations, including the non-rigorous selection method used for predictor selection and the limited clinical applicability of the developed models. Therefore, it becomes crucial to identify clinically meaningful and cost-effective prognostic factors that are readily available at the time of BM onset, as this would provide more valuable insights. This study aimed to establish a novel prognostic model based on clinicopathological characteristics, serological indicators, and treatment information using least absolute shrinkage and selection operator (LASSO)-Cox regression analysis to bridge this knowledge gap, thereby achieving a more precise reflection of the prognostic information on NSCLC patients diagnosed with BM. Our model could assist clinicians in formulating reasonable treatment plans.

## MATERIALS & METHODS

This clinical prediction model was constructed according to the transparent reporting of a multivariable prediction model for individual prognosis or diagnosis (TRIPOD) checklist (*Collins et al., 2015*).

This research adheres to the Declaration of Helsinki. The Ethics Committee of the Third Affiliated Hospital of Kunming Medical University approved this study (review number: KYLX2022221). Given the retrospective design of the study and the challenges associated with assessing certain patients, the Ethics Committee granted a waiver of informed consent for a subset of the patients.

### Study population and follow-up

This retrospective study included 300 patients with NSCLC who were diagnosed with BM between January 2006 and May 2020 at Yunnan Cancer Hospital, Third Affiliated Hospital of Kunming Medical University. The inclusion criteria for this study were as follows: (1) NSCLC confirmed *via* pathological examination; (2) magnetic resonance imaging-confirmed BM; (3) availability of patient demographic characteristics, clinicopathological features, serological indicators, and treatment information; (4) absence of other concurrent cancers. The survival duration of the patients was determined by reviewing the medical records and conducting telephonic inquiries. The OS was defined as the interval from the initial diagnosis until death from any cause or the date of the last follow-up visit (*Pilz, Manegold & Schmid-Bindert, 2012*).

### Data collection

Baseline clinical data were obtained from medical records at the time of initial diagnosis of BM. The collected data encompassed various aspects, including general conditions (age, sex,

body mass index (BMI), smoking history, and KPS), tumour markers (carcinoembryonic antigen (CEA), neuron-specific enolase (NSE), cytokeratins (cytoplasmic protein fragment of cytokeratin 19 (CYFRA21)), and squamous cell carcinoma antigen (SCCA)), serological indicators (albumin (ALB), lactate dehydrogenase, and alkaline phosphatase (ALP)), serum inflammatory indicators (neutrophils, platelets, lymphocytes, monocytes, platelet/lymphocyte ratio (PLR), neutrophil/lymphocyte ratio (NLR), systemic immune-inflammation index (SII) = platelet × neutrophil/lymphocyte, advance lung cancer inflammation index (ALI) = BMI × ALB/NLR), prognostic nutritional index (PNI) = ALB + 5× lymphocyte), advance distant metastases (number of BM, lung metastasis, intrathoracic metastasis (malignant pleural effusion, pericardial effusion, or pleural metastasis), liver metastasis, bone metastasis, adrenal metastasis, and metastases to other sites), signs and symptoms of BM (intracranial hypertension, focal signs and symptoms, epilepsy, and decreased cognitive function), type of pathology, pathological stage (tumour stage, lymph node stage (N_stage), or metastasis stage (M_stage)/TNM stage), epidermal growth factor receptor (EGFR) gene mutation status, treatment status (surgery for primary lung cancer foci, radiotherapy of primary lung cancer, radiotherapy for BM lesions (whole-brain radiation therapy or stereotactic radiation therapy), surgical treatment of metastatic brain lesions, chemotherapy, or EGFR-tyrosine kinase inhibitor (TKI) treatment), and classification information of RPA, GPA, Lung-molGPA, and BSBM models. The above predictors were complete and comprised objective data. All predictors were assessed independently of each other, without any knowledge of the clinical outcome. All continuity predictors maintained their continuity and were not categorised. The categorised predictors were all predetermined before model construction. The continuous and categorical variables are presented in Table S1. The sample size of this study met the criterion of having events per variable (EPV) of >10 (*Peduzzi et al., 1996*; *Austin, Allignol & Fine, 2017*; *Moons et al., 2019*).

## Model construction and evaluation

The final predictors were selected *via* a 10-fold cross-validation of LASSO-Cox regression, whereby the λ-value associated with the minimum standard error was chosen. Subsequently, the final predictors were incorporated into a multivariate Cox regression analysis, and the risk score for each patient was calculated using the "predict ()" function. Finally, a prognostic model was constructed.

*Risk score* $= h_0(t) \times \exp(\beta_1 X_1 + \beta_2 X_2 + \cdots + \beta_n X_n)$

The discriminatory ability of the model was assessed by evaluating the area under the receiver operating characteristic (ROC) curve (AUC). Furthermore, the calibration curve was plotted and the Brier score was calculated to measure the calibration of the model. Internal validation was conducted using the bootstrap method (resampling 1,000 times). The discriminative ability and clinical utility of the novel prognostic models were compared with RPA, GPA, Lung-molGPA, BSBM, and TNM staging using time-dependent C-index and decision curve analysis (DCA). A larger AUC value indicated better predictability of the model (*Carrington et al., 2023*). DCA demonstrated the relationship between benefits

and risks across models by examining various cut points (thresholds) in different models (*Van Calster et al., 2018*). Furthermore, the integrated discrimination improvement (IDI) and net reclassification improvement (NRI) were employed to assess the reclassification performance and discrimination of our novel prediction models compared with RPA, GPA, Lung-molGPA, BSBM, and TNM staging. A nomogram was developed based on the selected predictors to facilitate individual survival prediction in NSCLC patients with BM. Subsequently, patients were classified into low-risk, intermediate-risk, and high-risk groups based on the new prediction model RiskScore. The differences in OS among these three subgroups were analysed using the Kaplan–Meier method. All statistical analyses were performed using R software (version 4.2.1; *R Core Team, 2022*), and statistical significance was set at $P \leq 0.05$.

## RESULTS

### Patient characteristics

A total of 300 NSCLC patients with BM were included in this study, all of whom had complete baseline clinical and laboratory data. The clinicopathological characteristics and laboratory results of these patients are summarised in Table S1. The mean age of the patients was 55.4 years (range 31–83 years). Among the patients, 185 were males and 115 were females. The median follow-up duration of the patients was 13.9 months, with a minimum and maximum follow-up duration of 0.1 months and 173.83 months, respectively. The last follow-up was performed on 16 June 2021. The OS rates for these patients at 1, 3, and 5 years were 75%, 49%, and 40.3%, respectively.

### Construction of the prognostic models

First, LASSO-Cox regression analysis was used to identify the optimal predictors and constructed the model. Cross-validation was used to select the λ-value associated with lambda.min ($λ = 0.054$), which yielded the highest model fit (Fig. 1). This λ-value corresponds to the most significant prognostic factor for OS. The final model comprised 15 predictors, namely age, KPS, NSE, PLR, lymphocyte count, ALP, smoking history, intrathoracic metastasis, metastases to other sites, N_stage, M_stage, surgery for primary lung cancer foci, chemotherapy EGFR mutation, and TKI treatment. The EPV for each variable was 12.4. The regression coefficients of these predictors were used to construct the prognostic model. The risk score of the prognostic model was calculated as follows: risk score = h0(t) exp [(age × 0.0105543) + (kps × −0.0082627) + (NSE × 0.0017274) + (PLR × 0.0007071) + (lymphocyte count × −0.0174690) + (ALP × −0.0001577) + (smoke × 0.1433414) + (intrathoracic_metastasis × 0.4382260) + (metastases_to_other_sites × 0.0330467) + (N_stage × 0.1062274) + (M_stage × 0.0277756) + (surgery × −0.2624173) + (chemotherapy × −0.0027364) + (TKI_therapy × −0.3508606) + (EGFR mutation × −0.0952148)]. The continuous variables in the formula were based on the original numerical levels, while the categorical variables were represented by codes, as presented in Table S1.

We conducted Cox proportional risk regression analysis using the 15 predictors identified through LASSO regression (Fig. 2). The time-dependent ROC curves suggested AUC values

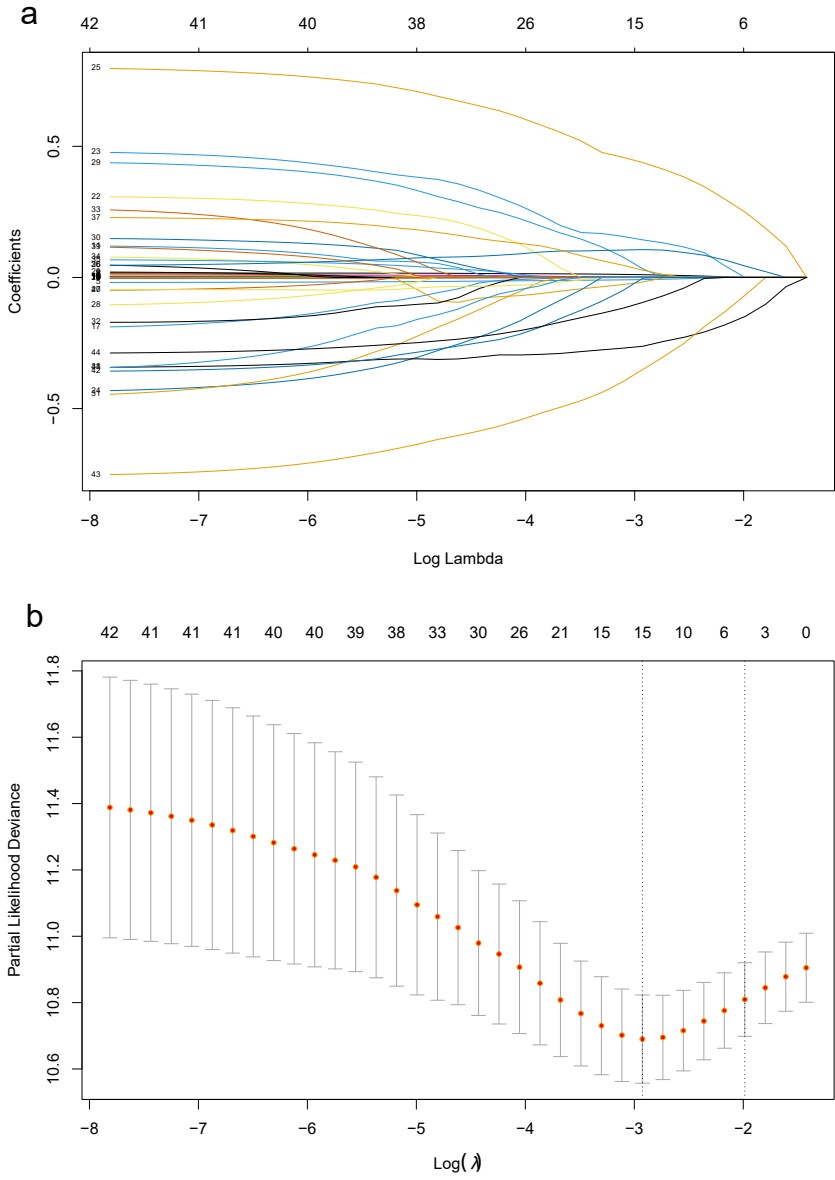

**Figure 1** **The LASSO regression algorithm was used for screening the predictors.** The LASSO regression algorithm was used for screening the predictors. (A) Path diagram of regression coefficients. Each curve represents the trajectory of each independent variable coefficient with log(ë). (B) Cross-validation curves of the LASSO regression analysis. The left dashed line is lambda.min, which is the smallest deviation of ë, while the right dashed line is lambda.1se, which is one standard error to the right of the smallest ë.

of 0.746 (0.678−0.814), 0.819 (0.761−0.877), and 0.865 (0.774−0.957) for our prediction model at 1, 3, and 5 years, respectively. The Brier score was calculated to measure the overall performance of the model, yielding a value of 0.103 (0.072−0.134). The Brier score combines discrimination and calibration measures, with lower values indicating better model performance (0 for perfect overall performance and 0.25 for worthless model)

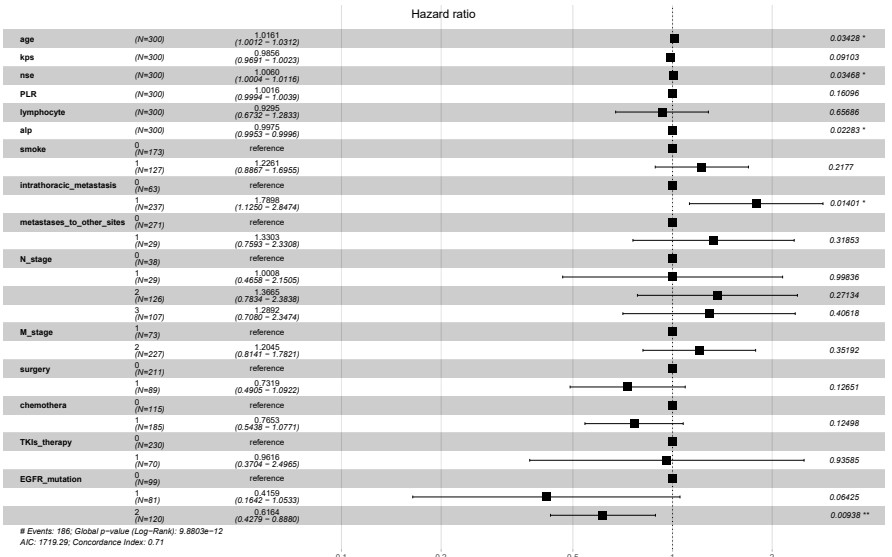

**Figure 2** **The hazard ratios and 95% confidence intervals for the 15 predictors.**

(*Schneider et al., 2022*; *Assel, Sjoberg & Vickers, 2017*). Additionally, internal validation using the bootstrap method (resampling of 1,000) demonstrated an AUC of 0.811 (0.638−0.950) and a Brier score of 0.123 (0.066−0.188; based on 5 years), confirming the model's good discriminative ability and calibration (Fig. 3).

## Evaluation of the performance between our novel prognostic model, RPA, GPA, Lung-molGPA, BSBM, and TNM staging

The time-dependent C-index was used to evaluate the accuracy of our model's predictions. Our model demonstrated superior discriminatory ability compared with RPA, GPA, Lung-molGPA, BSBM, and TNM staging, and these results were consistent with our bootstrap validation (Fig. 4).

Furthermore, DCA was employed to evaluate which focus the clinical applicability of our predictive model. Notably, the applicable threshold range varied among the six DCA curves. The widest threshold range was observed for our model, approximately 0.4−0.8, among the six curves. Moreover, within most of the threshold ranges, our model yielded the highest net benefits (Fig. 5). These findings indicate that our model is the optimal choice.

Lastly, the IDI and NRI metrics were incorporated to assess the performance of our model. The IDI analysis allowed us to measure the extent to which our new model improved its predictive power compared with the existing model, and the NRI analysis was used to determine the extent to which the new model improved the proportion of correct reclassifications compared with the existing model. The IDI analysis revealed that our model demonstrated positive improvements compared with GPA 0.152 (0.063−0.287), RPA 0.209 (0.113−0.347), Lung-molGPA 0.106 (0.030−0.240), BSBM 0.120 (0.044−0.247), and TNM staging 0.218 (0.122−0.354) with IDI >0. Meanwhile, the NRI analysis
a

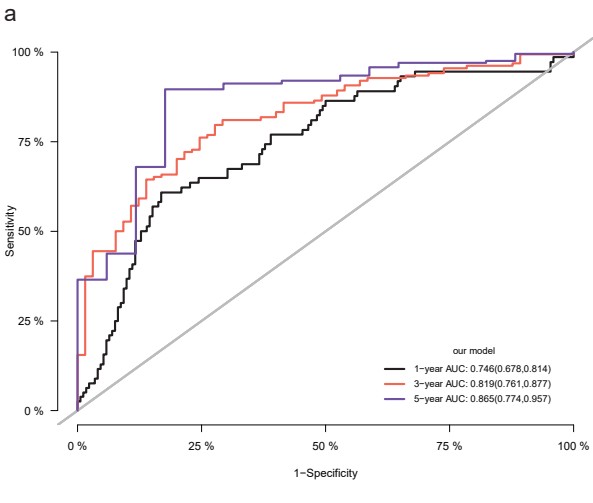

b

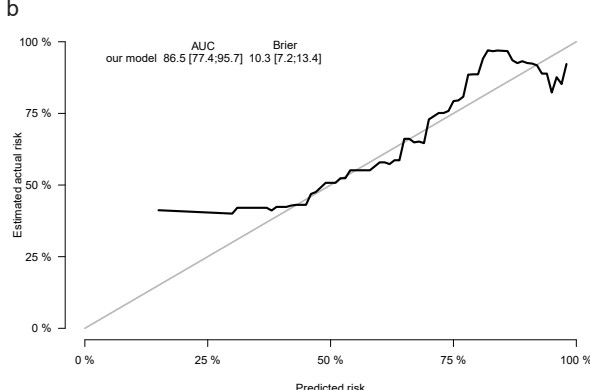

c

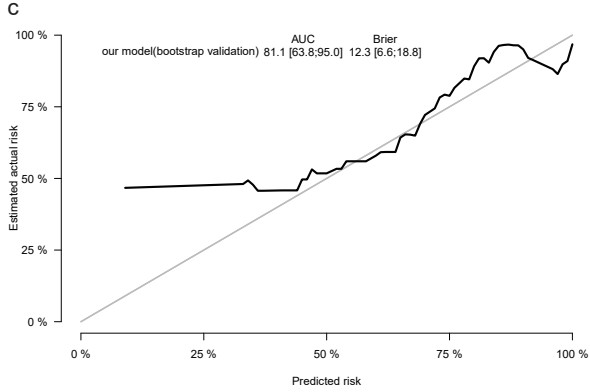

**Figure 3   The ROC, calibration curves, and Brier scores for the prediction model and internal valida-tion.** (A) The time-dependent ROC curves, with AUC values and 95% confidence intervals, of the prediction model for 1, 3, and 5 years; (B) calibration curves, with AUC values, Brier scores, and 95% confidence intervals, based on 5 years. The solid gray line represents a perfect prediction of an ideal model, while the solid black line indicates the performance of the constructed model. (C) The internal validation using the bootstrap method (resampling of 1,000).

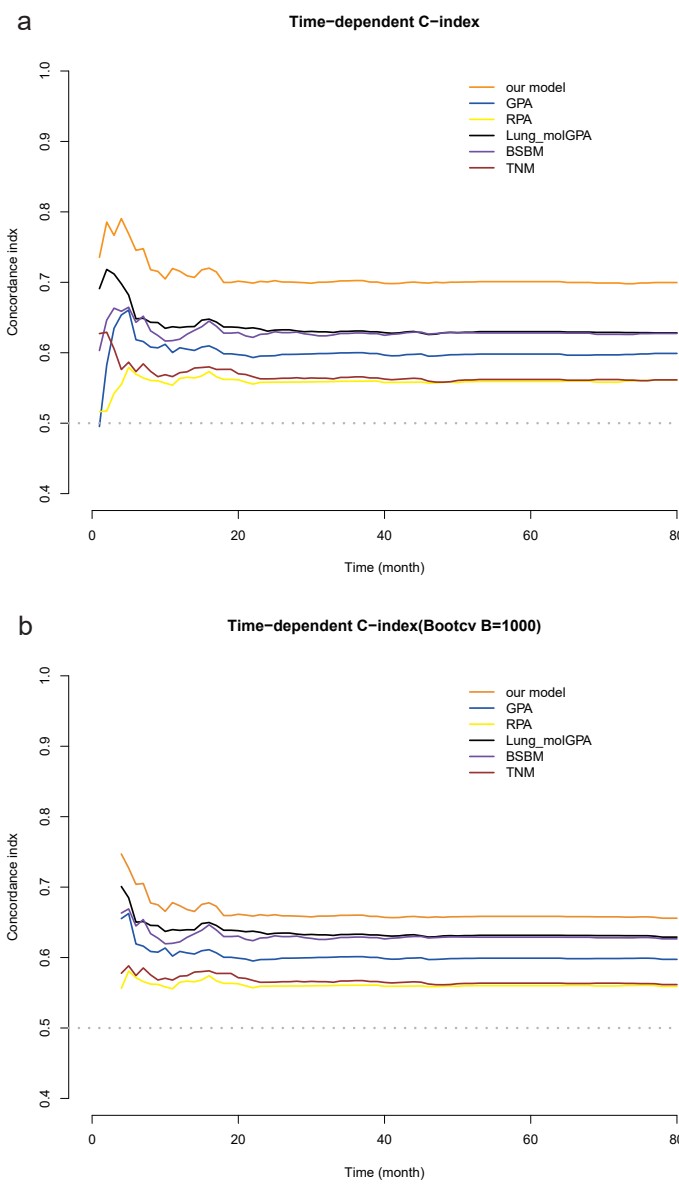

**Figure 4** **Comparison of the time-dependent C-index of the six models.** (A) Comparison of our models with RPA, GPA, Lung-molGPA, BSBM and TNM staging based on the time-dependent C-indices. (B) Internal validation using the bootstrap method.

revealed that our model demonstrated positive improvements compared with GPA 0.537 (0.172−0.676), RPA 0.474 (0.270−0.722), Lung-molGPA 0.525 (0.124−0.641), BSBM 0.457 (0.149−0.644), and TNM staging 0.536 (0.269−0.711) with NRI >0 (Table 1). These results indicate that our new model has effectively improved the outcome events compared with the previous model.

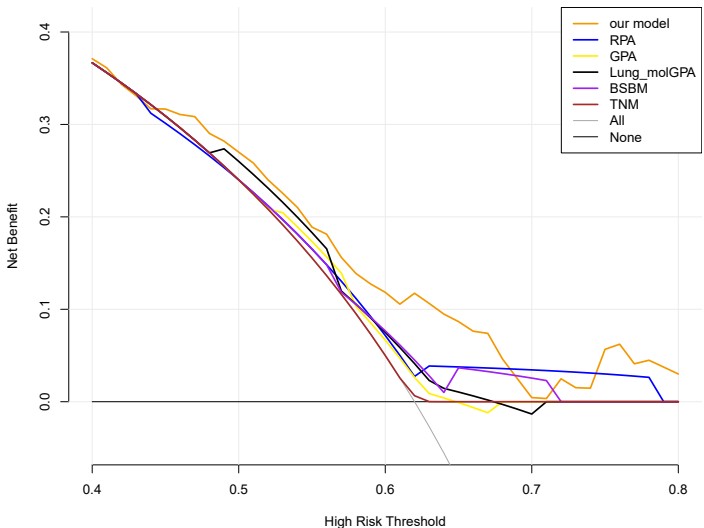

**Figure 5  The DCA curves of the six models.** The decision curves show that the threshold probabilities of our models range from 0.4−0.8, which is the widest threshold range of all models. Among most of the threshold ranges, our constructed model has the highest net benefit overall DCA curves compared to the curves for all treatment ("ALL" curve), no treatment ("None" curve), and the other three models.

**Table 1  IDI and NRI were used to evaluate the improvement in predictive power and proportion of correct reclassifications of our model compared to the older models for RPA, GPA, Lung-molGPA, BSBM and TNM staging.**

|  | IDI* (95% CI) | P value | NRI* (95% CI) | P value |
|---|---|---|---|---|
| our model *vs* GPA | 0.152(0.063~0.287) | 0.002 | 0.537(0.172~0.676) | 0.002 |
| our model *vs* RPA | 0.209(0.113~0.347) | <0.001 | 0.474(0.270~0.722) | 0.002 |
| our model *vs* Lung_molGPA | 0.106(0.030~0.240) | 0.014 | 0.525(0.124~0.641) | 0.01 |
| our model *vs* BSBM | 0.120(0.044~0.247) | 0.002 | 0.457(0.149~0.644) | 0.004 |
| our model *vs* TNM | 0.218(0.122~0.354) | <0.001 | 0.536(0.269~0.711) | <0.001 |

**Notes.**
IDI, Integrated Discrimination Improvement; NRI, Net Reclassification Index.
*Positive value represents better accuracy, negative velue represents worse accuracy.

## Constructing a nomogram for predicting OS

A nomogram was constructed to visualise our model. This provided a convenient, personalised tool to predict the 1-, 3-, and 5-year OS in NSCLC patients with BM. Each predictor is associated with a score, and the scores of all predictors were summed together to obtain a total score, from which the OS at 1, 3, and 5 years could be obtained (Fig. 6).

## Risk stratification based on our model

The risk score of each patient was calculated, following which they were classified into three groups, namely the low-risk ($n = 75$), intermediate-risk ($n = 150$), and high-risk ($n = 75$) groups based on the tertiles of their risk score to assess whether our model could accurately assess patient risk. The OS was significantly lower ($P < 0.001$) in the high-risk group (risk

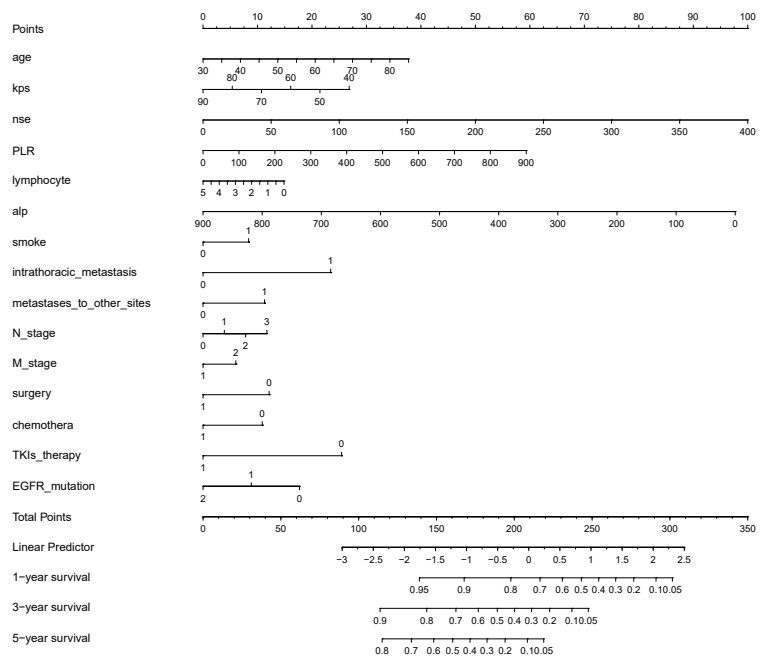

**Figure 6** **Constructed nomogram for predicting 1-, 3-, and 5-year OS in NSCLC patients diagnosed with BM based on 15 predictors.** The nomogram is used by summing the points for each prognostic factor. The total score on the bottom scale corresponds to the patient's probability of survival at 1, 3, and 5 years.

score >2.125) compared with the low-risk (risk score ≤0.789) and intermediate-risk (0.789 < risk score ≤2.125) groups. Additionally, patients in the intermediate-risk group had a lower OS than those in the low-risk group ($P < 0.001$).

The Lung-molGPA and BSBM models, along with our model, help distinguish the OS of patients. However, when using risk groupings derived from the GPA and RPA models, the differentiation of OS among patients is not fully effective. Specifically, according to the GPA model, there was no statistically significant difference in the OS between patients in the "GPA 1.5−2.5" group and the "GPA 3" group ($P = 0.275$). Similarly, based on the RPA model, there was no statistically significant difference between the OS of patients in the "class II" group and the "class III" group ($P = 0.122$).

While the dichotomous risk grouping based on TNM staging can differentiate patients' OS, it lacks the level of detail provided by our model, which comprises three subgroups. The results demonstrate the superior performance of our prognostic model in accurately distinguishing the prognosis of NSCLC patients with BM (Figs. 7, 8).

## DISCUSSION

In this study, a prediction model based on the LASSO-Cox regression algorithm was developed for predicting OS in NSCLC patients with BM. The prediction model comprised 15 variables, including age, KPS, NSE, PLR, lymphocyte count, ALP, smoking history, intrathoracic metastasis, metastases to other sites, N_stage, M_stage, surgery for primary

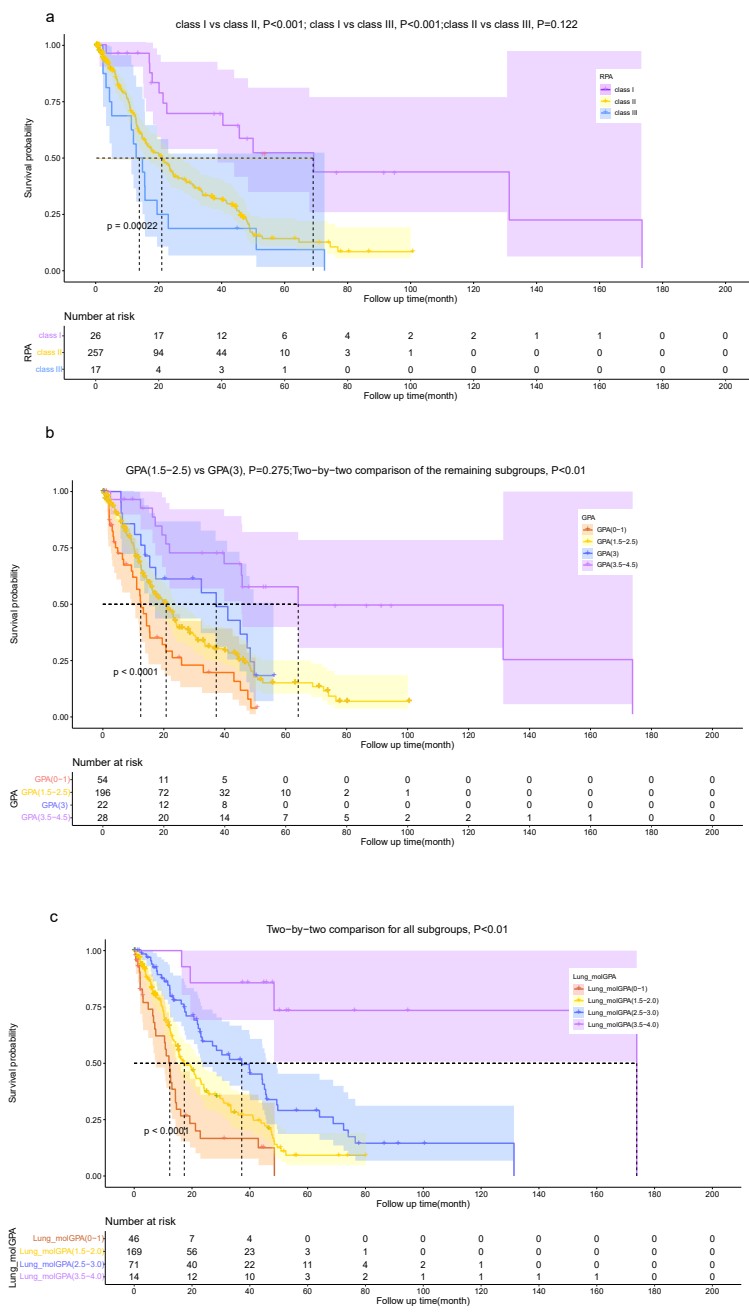

**Figure 7   Kaplan–Meier survival curve analysis for different models.** Kaplan–Meier plots are shown for RPA (A), GPA (B), Lung-molGPA (C), BSBM (D), our model (E) and TNM staging (F).

lung cancer foci, chemotherapy, EGFR mutation, and TKI treatment. The prediction model exhibited good discriminative ability, calibration, and clinical utility. In addition, it outperformed the conventional BM models such as GPA, RPA, Lung-molGPA, BSBM, and TNM staging. Furthermore, the patients were categorised based on their risk scores,

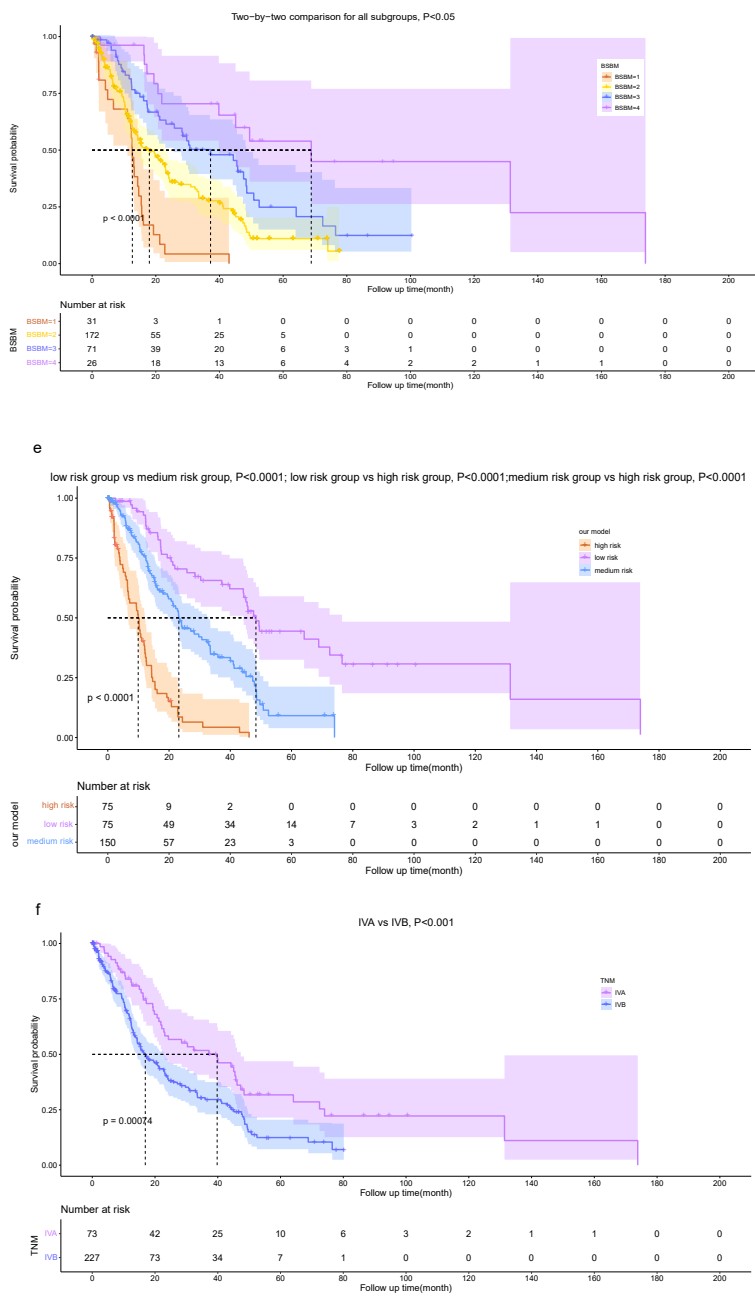

**Figure 8   Kaplan–Meier survival curve analysis for different models.** Kaplan–Meier plots are shown for RPA (A), GPA (B), Lung-molGPA (C), BSBM (D), our model (E) and TNM staging (F).

demonstrating significant differences in the OS of the low-, intermediate-, and high-risk subgroups as classified by our model.

Previous studies have identified that age, surgical treatment of the primary tumour, KPS, extracerebral metastasis, targeted therapy, NSE level, ALP level, and PLR are factors that

influence the prognosis of NSCLC patients with BM (*Rodrigus, de Brouwer & Raaymakers, 2001*; *Sanchez de Cos et al., 2009*; *Fuchs et al., 2021*; *Junger et al., 2021*; *Yu et al., 2021*; *Jacot et al., 2001*; *Cho et al., 2021*). In our model, five individual prognostic factors, namely NSE, PLR, ALP, intrathoracic metastasis, and targeted therapy, were identified. Notably, these factors were not considered in previously published clinical prediction models for NSCLC with BM. For instance, *Jacot et al. (2001)* demonstrated that high serum NSE levels were associated with a worse prognosis in NSCLC patients with BM, suggesting a correlation between elevated NSE and the extent of tumour-induced damage to normal brain tissue. On the other hand, PLR serves as an index of inflammation, and a study by *Cho et al. (2021)* reported that an increase of 10 in PLR was associated with a 1.3% increase in the risk of death in NSCLC patients with BM. These findings might be attributed to the association between inflammation and cancer progression, wherein elevated platelet levels result in the production of inflammatory cytokines and chemokines, thereby facilitating tumour progression (*Lim et al., 2019*). Additionally, lymphocytes play a crucial role in antitumour immunity, and a decrease in lymphocyte count indicates an impaired cell-mediated immune response and compromised antitumour immunity (*Jiang et al., 2019*). *Jacot et al. (2001)* observed that NSCLC patients with BM and elevated ALP levels had shorter survival. Moreover, the use of TKIs targeting driver mutations in NSCLC, such as EGFR-TKIs and anaplastic lymphoma receptor tyrosine kinase gene (ALK)-TKIs, has significantly improved the prognosis of NSCLC patients with BM who possess corresponding gene mutations (*Rotow & Bivona, 2017*; *Planchard et al., 2018*). Based on the above evidence, the predictors incorporated into our model are valid and plausible. However, it is worth noting that there is a controversial point regarding intrathoracic metastasis. While the study by *Hirashima et al. (2014)* found intrathoracic metastasis to be a significant favourable prognostic factor for NSCLC patients with distant metastases, our study arrived at the opposite conclusion, suggesting that intrathoracic metastasis is an unfavourable prognostic factor in NSCLC patients with BM. Further research is warranted to validate these findings and address the existing conflicts.

Few studies have developed predictive models for assessing the prognostic risk of NSCLC patients with BM, and these studies have certain limitations. For instance, *Wang et al. (2021)* constructed clinical prediction models using univariate analysis to screen predictors (*Zhang et al., 2020*; *Wang et al., 2021*; *Huang et al., 2020*). However, according to the prediction model risk of bias assessment tool guidelines, bias can arise when univariate analysis leads to the exclusion of certain variables from the, as some predictors only demonstrate significance when adjusted for other factors simultaneously during analysis (*Moons et al., 2019*). In contrast, our model is based on LASSO regression, which effectively filters predictors. This method is a robust reduction algorithm that actively selects relevant and interpretable predictors from a large pool of variables, considering possible multicollinearity. Moreover, it helps avoid model overfitting (*Balachandran et al., 2015*; *McEligot et al., 2020*). Another study by *Zhang et al. (2020)* constructed a predictive model based on a predictor known as the computed tomography imaging histology score (Rad-score). However, the clinical applicability of their model is limited since the Rad-score predictor is not readily available during hospitalisation as it is not a routine examination

item (*Zhang et al., 2020*). In contrast, our model incorporates readily available predictors, enhancing its clinical applicability. Similarly, *Li et al. (2022)* also established a nomogram combining patient clinicopathological factors and serological inflammatory markers (PLR, NLR, SII, PNI, and ALI) to predict survival in NSCLC patients with BM. Our study not only includes these indices but also incorporates a range of lung cancer-related tumour markers, such as CEA, NSE, CYFRA21, and SCCA, which have a potential impact on the prognosis of NSCLC patients with BM.

In summary, our LASSO-Cox regression analysis revealed that our prediction model provides a good fit for predicting OS in NSCLC patients with BM. The calibration plots demonstrated good calibration, while the time-dependent C-index analysis confirmed the model's strong prognostic accuracy compared with RPA, GPA, Lung-molGPA, BSBM, and TNM staging. Additionally, DCA revealed that our model yields the highest overall net benefit. Moreover, the IDI and NRI results showed significant improvements in predictive power and reclassification ratio when compared with RPA, GPA, Lung-molGPA, BSBM, and TNM staging. Notably, our model effectively stratified NSCLC patients with BM into low-, intermediate-, and high-risk subgroups, with the high-risk group exhibiting the worst survival outcomes. In conclusion, our clinical prediction model offers numerous advantages, including cost-effectiveness, broad applicability, simplicity of use, accessibility, and high accuracy. It holds great potential for predicting prognosis and aiding treatment decisions in NSCLC patients with BM.

Our study has certain limitations. First, the retrospective nature of our study introduces the possibility of selection bias, information bias, and confounding bias. Second, the data were obtained from a single hospital, and the sample size was relatively small. Therefore, future studies incorporating larger, multicentre cohorts are warranted to validate our findings. Furthermore, the inclusion of 15 predictive factors in our model might limit its clinical applicability due to the complexity associated with a large number of predictors. Finally, while our model incorporates easily accessible predictors, it is important to recognise that the specificity of the prognostic model could be improved by including NSCLC-related immunohistochemical markers or other relevant genetic mutations, such as programmed death-ligand 1, cytotoxic T-lymphocyte–associated antigen 4, ALK rearrangement, and ROS1 rearrangement (*Ahmadzada et al., 2018*).

## CONCLUSIONS

The clinical prediction model we constructed holds the potential for predicting OS in NSCLC patients with BM, outperforming established models such as RPA, GPA, Lung-molGPA, BSBM, and TNM staging.

## ACKNOWLEDGEMENTS

We express our appreciation to Bullet Edits for their expert language services.

### Funding
The authors received no funding for this work.

### Competing Interests
The authors declare there are no competing interests.

### Author Contributions

- Fei Hou conceived and designed the experiments, performed the experiments, analyzed the data, prepared figures and/or tables, authored or reviewed drafts of the article, and approved the final draft.
- Yan Hou performed the experiments, analyzed the data, prepared figures and/or tables, authored or reviewed drafts of the article, and approved the final draft.
- Xiao-Dan Sun performed the experiments, prepared figures and/or tables, and approved the final draft.
- Jia lv performed the experiments, prepared figures and/or tables, and approved the final draft.
- Hong-Mei Jiang analyzed the data, prepared figures and/or tables, and approved the final draft.
- Meng Zhang analyzed the data, prepared figures and/or tables, and approved the final draft.
- Chao Liu conceived and designed the experiments, authored or reviewed drafts of the article, and approved the final draft.
- Zhi-Yong Deng conceived and designed the experiments, authored or reviewed drafts of the article, and approved the final draft.

### Data Availability
The clinicopathological characteristics and laboratory results of these patients are available in the Supplementary File.

### Supplemental Information
Supplemental information for this article can be found online at http://dx.doi.org/10.7717/peerj.15678#supplemental-information.

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
