# Peer review of "Establishment of a prognostic risk prediction modelfor non-small cell lung cancer patients with brainmetastases: a retrospective study"

_PeerJ, doi:10.7717/peerj.15678_

## Round 0.1 · original submission · Major Revisions

Dear Dr. Hou,

Thank you for submitting your manuscript "Establishment of a prognostic risk prediction model for non-small cell lung cancer patients with brain metastasis" to PeerJ. We have now received reports from the reviewers, and after careful consideration internally, we have decided to invite a major revision of the manuscript.

As you will see from the reports copied below, the reviewers raise some concerns regarding the experimental validation and the English language for editing. We suggest that you address them, especially for English editing, with some professional editing services. Without substantial revisions, we will be unlikely to send the paper back for review.

If you feel that you are able to comprehensively address the reviewers’ concerns, please provide a point-by-point response to these comments along with your revision. Please show all changes in the manuscript text file with track changes or color highlighting. If you are unable to address specific reviewer requests or find any points invalid, please explain why in the point-by-point response.

Thanks

Abhishek Tyagi, PhD
Academic Editor,
PeerJ

Reviewer 1 ·

Basic reporting

This manuscript is overall well-written, but the English could be improved to make it easier for readers to follow and understand. I have a few suggestions as follows:

1. line 65-66, it is confusing when "account for" is used here. I would suggest the author change it to " TMN staging does not take other variables into consideration to predict the patient prognosis, such as age, gender and histology"

2. line 80: change to "are subjective or difficult to quantify"

3. line 87, given the context, it's better to use "However" instead of "In adition"

4. Figure1, information needs to be given about what the variables of each color curve are

5.

Experimental design

no comment

Validity of the findings

no comment

Additional comments

The aim of this manuscript is to provide a prediction model for evaluating the clinical outcome of NSCLC patients with brain metastasis, which is an interesting research topic and does warrant further investigation. The authors constructed a model based on clinical features and compared this new model with the previous golden/common scoring system, which is scientifically rational and profound. The methods of model construction and evaluation are overall well illustrated, and the visualization of the predicting OS helps to make the results more straightforward.

The risk stratification part validates the effectiveness of this new prediction model. I think this manuscript is suitable to publish after a few revisions according to above my comments.

Reviewer 2 ·

Basic reporting

- The model was well put into the context of the challenges involving risk stratification for patients with this disease, and the references used were appropriate and useful.
- Although I appreciate the use of an editorial service to help with the English language, I still had difficulty understanding the meaning of certain sentences.
o For example, the sentence from line 61 to 63 is unclear. The authors may want to say that patients with the same TNM stage may have different prognoses based on the anatomical sites of disease or other factors.
o Another example is the sentence at line 174-175.
- Some terms used were ambiguous or technically incorrect. Here are a few examples:
o Any form of death line 112 should be death from any cause.
o Lines 135-136 should state continuous and categorical variables.
o I don’t know what reduced predictors means at line 139.
o The use of “net profit” at line 206 appears inappropriate as this is not a cost-effectiveness analysis.
o Line 274: “risky prognostic factor” should be changed to associated with a poorer prognosis or unfavorable prognostic factor.
- The article is well structured. The figures used are relevant and enhance the reader’s understanding of the model the authors developed.
- The authors should describe the limitations of their model in more detail, including what “potential biases” encompasses at line 311. A major limitation of the model that is not mentioned is how it is a very complex mathematical formula that encompasses 15 predictors and as such is not simple to use in the clinic.

Experimental design

- This retrospective analysis aims to create a new prognostic model to inform clinical outcomes of patients with NSCLC metastatic to the brain, which remains an incompletely answered question of importance in the clinic. The research question was well defined in the manuscript.
- The methods of the study appear rigorous, are well described and reproducible.
- I would have liked the authors to describe how the decision was made of how to define low, intermediate and high risk in their model.

Validity of the findings

- The underlying data has been provided by the authors and appear robust overall.
- Before a model such as this one can be applied to research or clinical settings, larger multi-centered validation should be performed, but this was already mentioned by the authors.
- The conclusion is in keeping with the findings and appropriately conservative.

---

## Round 0.2 · accepted · Accept

Dear Dr. Hou,
Thank you for your submission to PeerJ.
I am writing to inform you that your manuscript, - Establishment of a prognostic risk prediction model for non-small cell lung cancer patients with brain metastases: a retrospective study," has been accepted for publication.

Congratulations, and thank you for your submission.

With kind regards,

Abhishek Tyagi
Academic Editor
PeerJ Life & Environment

Reviewer 1 ·

Basic reporting

after a few edits, the article is well-written and uses professional and clear English.

Experimental design

The study design and method is well defined and clearly explained.

Validity of the findings

The conclusions are supported by the findings and are statistically sound.